# Trends in Smoking Prevalence and Intensity between 2010 and 2018: Implications for Tobacco Control in China

**DOI:** 10.3390/ijerph19020670

**Published:** 2022-01-07

**Authors:** Guoting Zhang, Jiajia Zhan, Hongqiao Fu

**Affiliations:** 1Department of Health Policy and Management, School of Public Health, Peking University Health Science Center, Beijing 100191, China; gtzhang@pku.edu.cn; 2Department of Economics and Public Policy, Business School, Imperial College London, London SW7 2AZ, UK; j.zhan21@imperial.ac.uk

**Keywords:** smoking prevalence, smoking intensity, tobacco control policies, China

## Abstract

Background: China is the world’s largest producer and consumer of cigarettes. Since 2010, the Chinese government has implemented many policies to combat the tobacco epidemic, yet little is known about their overall impacts. This study aims to investigate the trends in smoking prevalence and intensity between 2010 and 2018. Methods: We use five waves of data from China Family Panel Studies (CFPS), a nationally representative survey, to examine the trends in smoking prevalence and intensity. We use the chi-square test and *t*-test to examine differences across waves. Binary logistic regressions and linear regressions are applied to examine the association between smoking behaviors and risk factors. Results: The current smoking prevalence dropped from 30.30% in 2010 (90% CI 29.47–31.31) to 28.69% (90% CI 27.69–29.69) in 2018. As for smoking intensity, the average daily cigarettes consumption decreased steadily from 16.96 cigarettes (90% CI 16.55–17.36) in 2010 to 15.12 cigarettes (90% CI 15.07–15.94) in 2018. Smoking risk factors for men included marriage status, education level, employment status, alcohol consumption, and physical activities. The smoking risk was higher for women with a lower education level, lower household income, unemployment status, and alcohol consumption behavior. Conclusions: Our study shows declined trends in both smoking prevalence and intensity between 2010 and 2018, suggesting some positive progress in tobacco control in China. Nonetheless, to achieve the goal of reducing smoking prevalence among people aged 15 and above to less than 20% by 2030, the Chinese government needs to take stronger anti-tobacco measures.

## 1. Introduction

Tobacco control continues to be a top priority in global health promotion. Tobacco use is the leading cause of preventable deaths around the world. It has killed more than five million people every year since 1990, and the corresponding economic burden is rising, especially in low- and middle-income countries [1]. Even though China ratified WHO Framework Convention on Tobacco Control (FCTC) in 2005, the pace of China’s tobacco control was slow. In 2010, China was ranked in the bottom 20% of countries for the implementation of FCTC compliance [2]. Among all countries, China was the largest manufacturer and consumer of cigarettes. Nearly 0.5 trillion cigarettes were produced in China in 1980 and the number rose to 2.38 trillion in 2010, accounting for almost half of global cigarettes production [3,4]. Moreover, only 1% of cigarettes manufactured in China were exported, with the rest being sold domestically [5]. Consequently, the disease burden associated with tobacco use in China was substantial, both in terms of the direct cost of related disease treatment and the indirect burdens from loss of productivity. According to a World Health Organization report, the direct loss due to treatment of tobacco-attributed illnesses reached ¥53 billion, amounting to 1.5% of China’s total national healthcare expenditures in 2014, and productivity loss due to tobacco-attributed illnesses reached ¥297 billion [6]. Tobacco control was a huge challenge in China’s health sector.

Since 2010, the Chinese government has placed a greater emphasis on tobacco control. A significant step was the 12th five-year plan in 2011, in which the Chinese government announced the full implementation of no smoking in public places [7]. Since then, many subnational governments had successively passed new or strengthened existing regulations on smoking in public places. In a striking move, on 28 November 2014, Beijing passed a smoking control ordinance that was strictly in compliance with the FCTC requirements: a ban on smoking in all indoor public and working places, together with the removal of all indoor smoking rooms. In 2015, the Chinese Ministry of Finance increased the tobacco consumption tax on the wholesale price of cigarettes from 5 percent to 11 percent. A specific excise tax of 0.10 RMB (US $0.017) was also imposed on cigarettes per pack [8]. In 2016, the Chinese government released the Healthy China 2030 plan. It set a clear goal of reducing the prevalence of smoking among people aged 15 and above to less than 20% by 2030 [9].

A number of studies have investigated the prevalence of smoking in China. However, most of these studies documented the prevalence of smoking only for a certain year or a certain district [10,11,12,13,14]. Some public repositories and organizations have already revealed some descriptive results about China’s smoking prevalence, but they mainly focus on the global comparison of tobacco use or just investigate a certain year’s situation [6,15,16]. Studies on trends in smoking prevalence tended to concentrate on the first ten years of the twenty-first century or earlier periods [17,18,19]. China’s tobacco control progress in the latest ten years has not been well documented. Therefore, our study aims to analyze the trends in smoking prevalence and intensity across China in recent years. This study may contribute to evaluating the effects of tobacco control policies since 2010 and provide some implications for future actions against the tobacco epidemic.

## 2. Methods

### 2.1. Data Sources and Study Population

We used data from China Family Panel Studies (CFPS) to assess the progress of China’s tobacco control between 2010 and 2018. The CFPS was a nationally representative longitudinal survey conducted by the Institute of Social Science Survey at Peking University [20]. It was a biennial survey launched in 2010 and had gained approval from the Institutional Review Board of Peking University (Approval IRB00001052-14010).

Up to now, the CFPS had five waves of publicly released datasets, covering the years 2010, 2012, 2014, 2016, and 2018. Each survey covered 25 provinces excluding Hong Kong, Macao, and Taiwan, as well as Xinjiang, Tibet, Qinghai, Inner Mongolia, Ningxia, and Hainan, which represented 94.5% of the total population in mainland China. The CFPS mainly conducted face-to-face interviews aided by phone interviews. In the 2010 baseline survey, the response rate was 74.9% and it finally successfully interviewed 14,960 households. In the following surveys, it successfully collected data from 13,453 households, 14,144 households, 14,763 households, and 14,218 households, and the response rates were 79.4%, 77.9%, 77.12%, and 69.3%, respectively. It collected extensive information on demographic characteristics, socioeconomic status, health-related behaviors and many other respects. We primarily used information about smoking behaviors, demographic characteristics, and some other health behaviors for this study. 

We incorporated those aged over 15 in our analysis because the questionnaire for children aged 15 and below did not include any questions on smoking behaviors. We further excluded samples that had missing values on variables presented in Table 1. As a result, a total of 149,257 observations (33,487 from 2010, 30,688 from 2012, 29,067 from 2014, 29,070 from 2014, 26,945 from 2018) were included for smoking prevalence analysis. Sample sizes by subgroups were also presented (Appendix A Table A1). When we analyzed smoking intensity, we only included samples who were current smokers. Hence, 43,046 observations (9905 from 2010, 8957 from 2012, 8239 from 2014, 8125 from 2016, 7820 from 2018) were included for analysis.

### 2.2. Indicators

The primary variables of interest in this study were self-reported smoking status and daily cigarette consumption. The variable current smoker was assessed by the following question: “Did you smoke cigarettes in the past month?”. Those who smoked cigarettes in the past month were assigned a value of 1; otherwise, it equaled 0. We used the average number of cigarettes smoked per day to measure smoking intensity. This variable was assessed using the question, “How many cigarettes do you smoke on average per day?”. Non-smokers did not need to answer this question. These questions remained unchanged across waves. The previous studies used the information from the CFPS to analyze smoking behaviors in China [21,22,23].

To identify the risk factors of smoking prevalence and intensity, two sets of explanatory variables were used. The first set were sociodemographic variables. We included age, square of age, gender (female, male), education (primary school or below, secondary or high school, college or above), marital status (unmarried, married or cohabited, divorced or widowed), employment status (unemployed, self-employed, employed by others), region (eastern, central, western), rural/urban status, and household income (lower than average, equal to or higher than average). The second set of explanatory variables were those that measure health behaviors. Current drinkers were defined as those who drank more than three times a week. People who did physical activities referred to those who did exercise at least once a week. 

### 2.3. Statistical Analysis

We conducted a descriptive analysis to examine the trends in smoking prevalence and intensity between 2010 and 2018. Given that the CFPS was a longitudinal survey and it oversampled in five “large provinces” (they are Shanghai, Liaoning, Henan, Gansu, and Guangdong), we conducted statistical analyses using the cross-sectional weights for each wave to ensure that our results were nationally representative [20]. We presented variable means with 90% confidence intervals (CIs) and used the chi-square test and *t*-test to examine the differences in results between survey years. 

Given substantial differences in social and economic development across different strata, we examined smoking prevalence and intensity by geographic regions and urban/rural status. We separated samples by education level and by gender to test the education gradient and gender differences in smoking behaviors. Age-specific (16–29, 30–39, 40–49, 50–59, 60–69, ≥70) trends in smoking prevalence and intensity were also evaluated.

We used binary logistic regression to investigate factors associated with current smoking behaviors using pooled data from five waves. Results were presented as odds ratios (ORs) with 90% CIs. The modified Poisson regression was also used as a supplement for its easier interpretation of results, which were presented as risk ratios (RRs) with 90% CIs. Linear regression models were used to assess the associations between the average number of daily cigarettes consumption and the risk factors. Results were presented as coefficients with 90% CIs. We included two sets of explanatory variables in the regression analysis. The first set were sociodemographic variables, and the second set were variables about individual health behaviors. Definitions of these variables were described in the previous subsection. In addition, we added dummies of the survey years and geographic regions to control time and district fixed effects. Due to the large gender differences in smoking prevalence and smoking intensity, the regression models were gender stratified. All statistical analyses were conducted in Stata SE 15.0. 

## 3. Results

### 3.1. Smoking Prevalence

Table 1 shows the trends of smoking prevalence between 2010 and 2018. The overall smoking prevalence decreased steadily from 30.39% (90% CI 29.47–31.31) in 2010 to 26.31% (90% CI 25.38–27.25) in 2016 and then increased slightly to 28.69% (90% CI 27.69–29.69) in 2018. Smoking was much more prevalent in men than in women. Specifically, in 2010, 56.76% (90% CI 55.33–58.19) of men in China were smokers. This proportion was much higher than that for women (3.21%, 90% CI 2.50–3.93). The large gender difference remained unchanged across survey years (55.00% in 2012, 51.94% in 2014, 48.57% in 2016 and 53.12% in 2018 vs. 2.93% in in 2012, 2.75% in 2014, 3.52% in 2016 and 3.02% in 2018), as shown in Table 1. During this period, the smoking prevalence for men decreased significantly by 3.6 percentage points (pp) between 2010 and 2018, while among women, the decrease was not statistically significant. 

We found statistically significant regional differences in smoking prevalence. In 2010, the weighted prevalence of smoking in the western region was 33.14% (90% CI 31.22–35.06), followed by a prevalence of 30.37% (90% CI 28.94–31.81) in the central region and 28.47% (90% CI 27.28–29.65) in the eastern region. In 2018, this regional pattern remained unchanged and the smoking prevalence in the western region kept at the highest level across the three regions. The smoking prevalence decreased significantly by 1.5 pp in the eastern region and 1.7 pp in the central region between 2010 and 2018. 

The smoking prevalence by urban/rural status is also shown in Table 1. Between 2010 and 2018, rural residents’ smoking prevalence was consistently higher than that of urban residents, though, for both groups, the share of current smokers declined. The prevalence for urban residents decreased from 28.52% (90% CI 27.40–29.64) in 2010 to 27.81% (90% CI 26.75–28.87) in 2018, and the prevalence for rural residents declined from 32.28% (90% CI 30.95–33.61) in 2010 to 30.05% (90% CI 28.59–31.51) in 2018. It indicates that rural residents aged 16 and above experienced a greater decrease in smoking prevalence than that of urban residents. The gap in smoking prevalence between urban and rural residents had narrowed from 2010 to 2018.

Smoking prevalence varied significantly among people with different educational levels. People with a college degree or above had a much lower smoking prevalence compared with those with secondary or high school education and below. Additionally, smoking prevalence in people with a college degree or above experienced the greatest reduction from 2010 to 2018. People with primary education or below also had a decline of 2.6 pp in smoking prevalence between 2010 and 2018. As for people with secondary or high school education, the prevalence remained unchanged from 2010 to 2018. 

We also examined smoking prevalence by age group. The relationship between smoking prevalence and age was not linear. Take the wave of 2010 as an example. The smoking prevalence was 22.69% (90% CI 21.46–24.47) among people aged 16–29 and reached 35.65% (90% CI 34.27–37.03) among people aged 50–59. For people aged 60–69 and the group aged over 70, their shares of current smokers were 31.73% (90% CI 30.09–33.37) and 26.00% (90% CI 23.52–28.48), respectively. There was a similar age pattern in other waves, with the highest smoking prevalence among people aged 50–59. Moreover, between 2010 and 2018, smoking rates for people aged 40–49 and aged 50–59 decreased by 4.7 pp and 3.5 pp, respectively. Other age groups also saw a decrease in prevalence over time, but the changes were not statistically significant. Figure 1 provides a more detailed age-specific analysis of smoking prevalence. We can observe inverted U curves of smoking prevalence by age among men respondents. There was a sharp increase from 11.72% in the population aged 16–20 to 27.28% in the population aged 21–25 in 2010, suggesting that many people started smoking at a young age. Additionally, we find that smoking prevalence among people aged 16–20 increased from 11.72% in 2010 to 14.29% in 2018. A different pattern of age-specific prevalence was shown for women. Despite the fact that the curves of smoking prevalence in women showed a slight upward trend as age increased, the proportion of women who smoked remained at a low level over the period.

Table 2 shows the results of gender-stratified binary logistic regression on smoking behavior using pooled data from waves of 2010, 2012, 2014, 2016, and 2018, in which the odds ratios and 90% CIs for each regression are reported. We found risk factors for smoking behavior differed by gender. For men, those who were married or divorced had a much higher smoking prevalence compared with unmarried men (married or cohabited: OR = 1.430; *p* < 0.01; divorced or widowed: OR = 1.943, *p* < 0.01). Men with higher education were less likely to be smokers (secondary or high school: OR = 0.822, *p* < 0.01; college or above: OR = 0.512, *p* < 0.01). Being employed increased the odds of being a smoker (self-employed: OR = 1.381, *p* < 0.01; employed by others: OR = 1.654, *p* < 0.01). The association between smoking prevalence and the level of household income was positive (equal to or higher than average: OR = 1.033, *p* < 0.1). Current drinking behavior was positively associated with smoking behavior (OR = 1.835, *p* < 0.01). Doing physical activities decreased the possibility of smoking (OR = 0.796, *p* < 0.01). 

For women, unlike men, those who were married were less likely to be smokers (married or cohabited: OR = 0.602, *p* < 0.01). In addition, employment status and smoking behavior were associated in different ways for women. Women with jobs were less likely to smoke (self-employed: OR = 0.847, *p* < 0.01; employed by others: OR = 0.749, *p* < 0.01). Women with a higher level of household income had lower smoking prevalence (equal to or higher than average: OR = 0.659, *p* < 0.01). Similar to that of men, a lower level of education, lower level of household income, and current drinking behavior increased the odds of being a smoker among women.

We also used modified Poisson regression to estimate the risk ratios because results interpretation was easier (Appendix A Table A2). There were small differences between results from the two methods and the findings remained unchanged. For men, the risk of smoking was increased by 21.9% among those who were married or cohabited relative to the unmarried men (married or cohabited: RR = 1.219; *p* < 0.01). The divorced or widowed men had a 37.5% increase in risk of being current smokers, compared to the unmarried men (divorced or widowed: RR = 1.375, *p* < 0.01). Current smoking behavior was 1.169 times more likely to occur among self-employed people than among the unemployed (self-employed: RR = 1.169, *p* < 0.01). People who were employed by others had a 26.2% increase in risk of being current smokers compared to the unemployed (employed by others: RR = 1.262, *p* < 0.01). Current drinkers had 1.255 times the risk of smoking compared to non-drinkers (RR = 1.255, *p* < 0.01). For women, the risk of smoking among the married or cohabited was reduced by 38.8% relative to the unmarried (married or cohabited: RR = 0. 612, *p* < 0.01). Compared to the employed, the women who were self-employed and employed by others had a 14.5% and 24.1% reduction in risk of being current smokers, respectively (self-employed: RR = 0. 855, *p* < 0.01; employed by others: RR = 0.759, *p* < 0.01). The risk of smoking was 32.6% less among women with a higher level of household income compared with women with a lower level of household income (RR = 0.674, *p* < 0.01). 

### 3.2. Smoking Intensity

Table 3 presents trends in smoking intensity among current smokers between 2010 and 2018. Among current smokers, the average number of daily cigarettes smoked decreased from 16.96 (90% CI 16.55–17.36) in 2010 to 15.12 (90% CI 15.07–15.94) in 2018, suggesting that current smokers had consumed fewer cigarettes over time. 

There was also a gender gap on the average number of daily cigarettes smoked. In each wave, men smokers consumed more cigarettes than women smokers. During the 2010–2018 period, the average daily cigarettes consumed by men smokers decreased by 1.90, while the average daily cigarettes consumed by women smokers remained largely unchanged. As for trends across regions, the western region had the highest number of average daily cigarettes in the beginning, but then it kept at the lowest level from 2012 to 2018. In other words, the western regions experienced the greatest reduction of average daily cigarettes between 2010 and 2018. The eastern and central regions also displayed decreasing trends in smoking intensity among current smokers. The average daily cigarettes smoked for urban and rural smokers reduced from 16.31 (90% CI 15.85–16.78) and 17.54 (90% CI 16.98–18.08) in 2010 to 15.04 (90% CI 14.60–15.49) and 15.23 (90% CI 14.72–15.73) in 2018, respectively. Throughout the years, the smoking intensity gap between urban and rural smokers had narrowed.

We also observed that smokers with primary education or below consumed the most cigarettes on average per day, and their cigarettes consumption per day only decreased by 1.07 from 2010 to 2018. Smokers with higher education consumed fewer cigarettes. Especially for smokers with a college degree or above, their average daily cigarettes smoked were consistently lower than other groups. Table 3 also describes the average consumption of cigarettes per day across age groups. Notably, in most waves, the respondents aged 50–59 had the most average daily consumption of cigarettes, compared to other age groups. It was consistent with the finding in the smoking prevalence. Moreover, smokers aged 30–39 experienced the greatest decrease over time, followed by smokers aged 40–49 and aged 16–29. Yet the reduction in smokers aged 50–59 and those aged over 70 was not statistically significant. The relationship between cigarette consumption per day and age groups was also presented as an inverted U curve for men smokers, as shown in Figure 2. This inverted U pattern was consistent across years. Due to limited observations in each age group in our sample, we did not present the results of average daily cigarettes along with age for women smokers.

Table 4 shows the results of gender-stratified regressions on the number of average daily cigarettes using pooled data from five waves. The regressions are based on the ordinary least squares model. For men, a higher education level was associated with less average daily cigarette consumption among smokers. Men having jobs smoked more cigarettes. Current drinkers were positively associated with more average daily cigarettes consumption, compared with non-drinkers. People who did physical activities consumed fewer cigarettes daily. For women, higher education level also mattered for women’s daily cigarettes consumption, and it had a greater influence on women than on men. Having jobs or not had limited impacts on women smokers’ daily cigarette consumption. Similar to men, women’s drinking habits were also associated with increased daily cigarettes consumption, and women smokers who had physical activities consumed fewer daily cigarettes on average.

## 4. Discussion

Our study examined the temporal trends in both smoking prevalence and intensity since the Chinese government started to place a greater emphasis on tobacco control. Both encouraging and discouraging findings emerged. Between 2010 and 2018, the overall smoking prevalence in China fell by 1.7 percentage points. This declining trend was consistent with the findings from China Global Adult Tobacco Survey (from 28.1% in 2010 to 26.6% in 2018), though it reported a slightly lower smoking prevalence [24,25]. The average daily cigarette consumption decreased by 1.84 as well. Women’s smoking prevalence kept much lower. However, it should be noted that the overall smoking prevalence remained high, at 28.69% in 2018. Smoking control for men faced a great challenge with its high smoking prevalence of more than 50%. Furthermore, the smoking prevalence in the young aged 16–20 years increased from 11.72% in 2010 to 14.29% in 2018. If the tobacco control progress continues at the current pace without decisive actions, it may be difficult for the China government to achieve the goal of reducing smoking prevalence to 20% among people aged 15 and above by 2030, which was set in the Health China 2030 Plan. 

The Chinese government can learn from other countries experiences and then take more effective measures to combat the tobacco epidemic, given that many developed and developing countries have seen significant reductions in smoking rates after taking effective measures to discourage tobacco use. In England, the proportion of adults who smoke cigarettes declined from 20.1% in 2010 to 17.1% in 2015 [26]. The United States also achieved a dramatic decline in smoking prevalence, dropping from 20.9% in 2005 to 15.1% in 2015 [27]. In Thailand, the smoking prevalence among people aged 15 years old or above decreased from 32.0% in 1991 to 19.1% in 2017 [28]. In 2014 across Brazil, approximately 15% of the population were smokers, down from 34.8% in 1989 [29]. 

One of the measures to accelerate the progress of tobacco control can be strict regulations on smoking in public places. In recent years, the pace of subnational tobacco control legislation on smoking in public places has accelerated. To date, 107 prefecture-level cities have passed or strengthened regulations on smoking in public places [30]. However, in some of these cities, indoor smoking is not completely prohibited. A more worrisome fact is that about two-thirds of Chinese cities have never imposed any regulations on smoking in public places. In light of the difficulty in effectively enforcing smoking bans in some cities, China urgently needs a comprehensive smoking ban legislation at the national level.

In addition, China can make warning labels on cigarette packs more prominent. At present, there are only text warnings (“Smoking is harmful to health”) on cigarettes package but most cigarettes in China are adorned with beautiful illustrations. Pictorial warnings on packages is proved to be a highly cost-effective and population-wide tobacco control strategy. As of 2014, at least 77 countries or jurisdictions introduced pictures of tobacco-related diseases on cigarette packs, including Canada, Brazil, Singapore, Australia, Thailand, India, and Jamaica [31]. Thailand now has the largest warnings in the world at 85% of the package front and back, surpassing Australia at 82.5% [31]. Many initiatives prove that pictorial warnings are better in communicating tobacco-attributed diseases [32], encouraging smoking cessation [33], and discouraging smoking initiation [34], compared to text-only messages. Therefore, pictorial warning images on cigarette packs may be one of the policy options for China’s anti-tobacco movement. 

Many studies have found solid evidence that increasing the tobacco tax is an effective measure to encourage smoking cessation and reduce smoking intensity [35,36]. In China, taxes on tobacco have been raised twice but the level of taxes is still low. The first tax increase happened in 2009, and the second increase happened in 2015 [8]. However, even after the 2015 tax adjustment, the price of a pack of cigarettes remains relatively low and affordable, with the cost of 12.84 RMB per pack on average in 2015 [37]. In contrast, developed countries imposed much higher tax rates on cigarettes. For example, in 2010, Australia imposed a 25% excise tax on tobacco without warnings. Then since 2013, a 12.5% annual increase of excise tax had been implemented [38]. Consequently, Australia now has the world’s highest cigarette price and the cost of a pack has risen to $A40 by 2020 [39]. However, excise taxes in China only account for about 39% of the final retail price [40], which is still far below the WHO’s recommended level of 75%. Hence, the Chinese government still has much room to further raise cigarettes taxes.

Of particular concern in our results was the substantial increase in smoking prevalence in the young aged 16–20, from 11.72% in 2010 to 14.29% in 2018, as shown in Figure 1. Consistently, according to a previous finding, people aged 15–24 experienced the greatest increase in the prevalence of smoking among all age groups [18]. It also showed that 77.9% of current smokers began smoking during their adolescence [18]. The reason could be that many adolescents and young adults underestimate the health risks of smoking and regard it as a way of socializing. Due to the addictive substances of nicotine, most smokers have great difficulty quitting smoking in later years. Tobacco control initiatives, therefore, need to pay more attention to adolescents and young people. The reality, however, is that although the Law on Protection of Minors in China prohibits the sale of tobacco products to minors and requires identification checks, it is not strictly enforced in practice and adolescents are able to get tobacco products with little difficulty in China. Hence, there is an urgent need in China for an enforceable law to stop adolescents from smoking.

This study also examines the risk factors of smoking behaviors. We found that drinking was strongly associated with the likelihood of smoking in both men and women. Many studies have identified cigarettes and alcohol as complementary products that the utilization of one item may lead to a rise in demand for the other [41,42]. The co-use of tobacco and alcohol may further harm the population’s health. Lower education is also associated with higher smoking prevalence. Many studies supported that exposure to higher education can improve personal health awareness and promote healthier behaviors [43,44]. The phenomenon of higher smoking prevalence in rural areas was perhaps because of lower education levels and less strict anti-smoking interventions. As for the relation between smoking behavior and employment status, gender differences existed. Women with jobs were less likely to smoke perhaps because they might pay more attention to their image since smoking is always considered a masculine behavior in China [45]. By contrast, men having jobs exhibited a higher prevalence of smoking behaviors than those without jobs. As is revealed by some public health academics, this could be due to the fact that men smokers can get some social benefits by sharing cigarettes [46]. Moreover, in China, refusing an offered cigarette may also be regarded as rude behavior and may cause exclusion from social groups [47]. Gender differences also existed in the relation between smoking behaviors and marital status. Married men were more likely to be smokers compared to unmarried men, while unmarried women had a higher prevalence of smoking. The difference could be stemmed from that the fact that married women usually take more responsibility for childcare and spend more time with children than married men in China. Considering the harmful impacts of smoking on children’s health, married women tend not to smoke. 

Our study also has some limitations. First, the latest wave of CFPS surveys was conducted three years ago so we are unable to test trends after 2018. Second, though this study shows a general downward trend, an increased smoking prevalence appears between 2016 and 2018. The reasons for the increase in this period are unclear. Third, this is a cross-sectional study that emphasizes the association between smoking and risk factors. We cannot draw conclusions regarding causality. Fourth, cigarettes smoking is not the only way to use tobacco in China, though the percentage of cigarette smokers among current tobacco smokers reached 96.7% in 2018 [25]. The consumption of many other tobacco products such as e-cigarettes has been on the rise [48]. Future studies may include the utilization of other tobacco products for analysis when their data is available.

## 5. Conclusions

This study shows declined trends in both smoking prevalence and intensity between 2010 and 2018 in China, suggesting that the Chinese government had made some positive progress in tobacco control over the years. Nevertheless, at the current pace, it may be difficult for China to achieve the goal of reducing the adult smoking prevalence to 20% by 2030. We recommend that the Chinese government needs to take more effective measures against the tobacco epidemic. 

## Figures and Tables

**Figure 1 ijerph-19-00670-f001:**
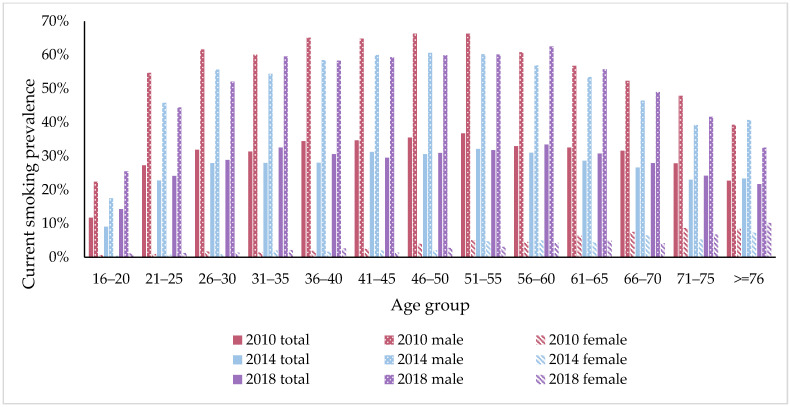
Age-specific prevalence of smoking prevalence in China in 2010, 2014 and 2018. Data source: China Family Panel Studies.

**Figure 2 ijerph-19-00670-f002:**
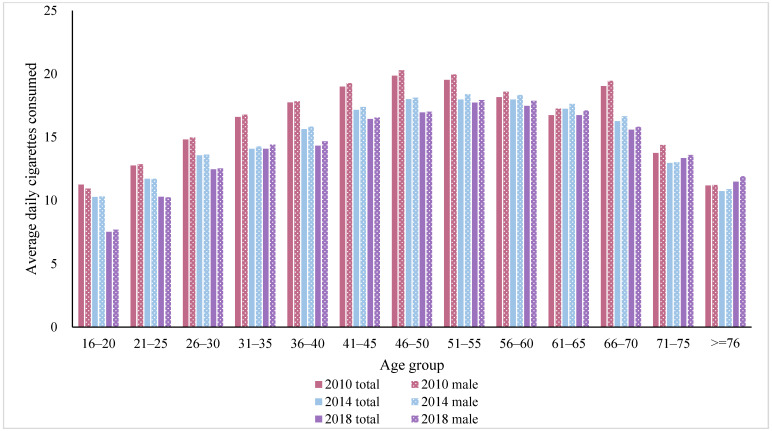
Age-specific trends of number of average cigarettes smoked among smokers in China in 2010, 2014 and 2018. Due to limited observations of women smokers in each age group in our sample, the results for women smokers were not presented.

**Table 1 ijerph-19-00670-t001:** Trends in smoking prevalence across region, residence, gender, educational level and age groups in China (2010–2018).

Current Smoker	2010	2012	2014	2016	2018	Difference		
	Weighted Proportion (90% CI)	Weighted Proportion (90% CI)	Weighted Proportion (90% CI)	Weighted Proportion (90% CI)	Weighted Proportion (90% CI)	2010–2014	2014–2018	2010–2018
Individuals	30.39% (29.47–31.31)	28.21% (27.34–29.08)	27.04% (25.94–28.14)	26.31% (25.38–27.25)	28.69% (27.69–29.69)	−0.033 ***	0.016 ***	−0.017 ***
**Gender**								
Female	3.21% (2.50–3.93)	2.93% (2.34–3.52)	2.75% (2.16–3.34)	3.52% (2.78–4.26)	3.02% (2.39–3.65)	−0.005 **	0.003	−0.002
Male	56.76% (55.33–58.19)	55.00% (53.52–56.47)	51.94% (49.98–53.90)	48.57% (47.07–50.06)	53.12% (51.44–54.79)	−0.048 ***	0.012	−0.036 ***
**Region**								
Eastern	28.47% (27.28–29.65)	27.26% (26.05–28.49)	24.80% (23.45–26.16)	25.17% (23.75–26.59)	26.94% (25.56–28.31)	−0.037 ***	0.021 ***	−0.015 **
Central	30.37% (28.94–31.81)	28.34% (26.70–29.98)	27.09% (26.12–28.06)	26.79% (25.24–28.34)	28.64% (27.20–30.07)	−0.033 ***	0.015 *	−0.017 *
Western	33.14% (31.22–35.06)	29.54% (27.77–31.32)	30.67% (28.59–32.75)	27.49% (25.72–29.25)	31.43% (29.17–33.69)	−0.025 **	0.008	−0.017
**Residence**								
Rural	32.28% (30.95–33.61)	29.10% (27.81–30.38)	28.89% (27.41–30.37)	28.26% (26.98–29.55)	30.05% (28.59–31.51)	−0.034 ***	0.012	−0.022 ***
Urban	28.52% (27.40–29.64)	27.42% (26.37–28.48)	25.80% (24.54–27.06)	25.04% (23.96–26.13)	27.81% (26.75–28.87)	−0.027 ***	0.020 ***	−0.007
**Education**								
Primary or below	30.48% (29.10–31.87)	27.93% (26.72–29.14)	27.44% (26.00–28.88)	26.43% (25.15–27.71)	27.90% (26.36–29.44)	−0.030 ***	0.005	−0.026 ***
Secondary or high school	31.09% (30.13–32.05)	29.62% (28.59–30.65)	27.68% (26.54–28.82)	27.53% (26.43–28.63)	31.54% (30.54–32.55)	−0.034 ***	0.039 ***	0.005
College or above	25.67% (23.59–27.75)	22.82% (20.72–24.93)	22.19% (19.87–24.52)	19.02% (16.91–21.13)	21.44% (19.40–23.49)	−0.035 **	−0.008	−0.042 **
**Age, years**								
16–29	22.96% (21.46–24.47)	20.05% (18.71–21.40)	19.74% (18.30–21.19)	17.20% (15.93–18.47)	22.22% (20.61–23.84)	−0.032 ***	0.025 **	−0.007
30–39	32.66% (31.22–34.10)	30.44% (28.83–32.05)	27.91% (26.09–29.74)	29.71% (27.99–31.43)	31.71% (29.77–33.64)	−0.048 ***	0.038 ***	−0.01
40–49	34.85% (33.48–36.22)	32.14% (30.75–33.52)	30.76% (29.12–32.41)	29.70% (28.09–31.31)	30.17% (28.50–31.84)	−0.041 ***	−0.006	−0.047 ***
50–59	35.65% (34.27–37.03)	33.92% (32.49–35.35)	31.46% (29.69–33.23)	30.67% (28.92–32.43)	32.15% (30.33–33.97)	−0.042 ***	0.007	−0.035 ***
60–69	31.73% (30.09–33.37)	29.70% (28.23–31.17)	28.56% (26.86–30.26)	29.11% (27.34–30.87)	29.88% (28.38–31.38)	−0.032 **	0.013	−0.019
≥70	26.00% (23.52–28.48)	23.61% (21.15–26.07)	23.46% (20.85–26.06)	22.19% (19.77–24.60)	24.21% (22.19–26.22)	−0.025	0.007	−0.018

Note: Weighted percentages and 90% confidence intervals were reported in the table. Chi-square test was used to compare the change in proportions. ***, ** and * denoted statistical significance at the 1%, 5%, and 10% level, respectively.

**Table 2 ijerph-19-00670-t002:** Risk factors associated with current smoking status.

	Total	Women	Men
Current Smoking	Odds Ratio (90% CI)	Odds Ratio (90% CI)	Odds Ratio (90% CI)
**Marital status**			
Unmarried	1(ref)	1(ref)	1(ref)
Married or cohabited	0.746 ***	0.602 ***	1.431 ***
	(0.714~0.779)	(0.475~0.764)	(1.360~1.505)
Divorced or widowed	0.697 ***	0.796	1.943 ***
	(0.657~0.739)	(0.618~1.026)	(1.802~2.096)
**Educational level**			
Primary or below	1(ref)	1(ref)	1(ref)
Secondary or high school	1.248 ***	0.720 ***	0.822 ***
	(1.218~1.278)	(0.656~0.790)	(0.798~0.847)
College or above	0.809 ***	0.523 ***	0.512 ***
	(0.774~0.846)	(0.413~0.663)	(0.486~0.540)
**Household income**			
Lower than average	1(ref)	1(ref)	1(ref)
Equal to or higher than average	0.909 ***	0.659 ***	1.033 *
	(0.889~0.930)	(0.606~0.716)	(1.004~1.064)
**Work type**			
Unemployed	1(ref)	1(ref)	1(ref)
Self employed	1.694 ***	0.847 ***	1.381 ***
	(1.646~1.744)	(0.774~0.927)	(1.330~1.435)
Employed by others	2.517 ***	0.749 ***	1.654 ***
	(2.440~2.597)	(0.660~0.852)	(1.590–1.722)
**Current drinker**			
No	1(ref)	1(ref)	1(ref)
Yes	5.141 ***	3.997 ***	1.835 ***
	(5.009~5.276)	(3.551~4.499)	(1.780~1.890)
**Physical activity**			
No	1(ref)	1(ref)	1(ref)
Yes	0.922 ***	0.954	0.796 ***
	(0.902~0.943)	(0.883~1.030)	(0.774~0.818)
**Observations**	146,694	74,457	72,237

Note: In columns 2 and 3, binary logistic regression models were adjusted for geographical regions, urban/rural status, age and age square, and wave dummies using the pooled data from waves of 2010, 2012, 2014, 2016, and 2018. In column 1, the binary logistic regression model further included the gender variable in addition to other control variables. 90% Confidence intervals for Odds Ratio were reported in the brackets. *** *p* < 0.01, * *p* < 0.1.

**Table 3 ijerph-19-00670-t003:** The weighted number of average daily cigarettes smoked among current smokers in China (2010–2018).

Number of Average Daily Cigarettes	2010	2012	2014	2016	2018	Difference		
	Weighted Means (90% CI)	Weighted Means (90% CI)	Weighted Means (90% CI)	Weighted Means (90% CI)	Weighted Means (90% CI)	2010–2014	2014–2018	2010–2018
Individuals	16.96 (16.55–17.36)	16.19 (15.78–16.60)	15.83 (15.42–16.24)	15.50 (15.07–15.94)	15.12 (14.76–15.48)	−1.125 ***	−0.710 ***	−1.836 ***
**Gender**								
Female	12.14 (11.27–13.01)	13.03 (12.09–13.96)	12.12 (10.97–13.28)	10.95 (9.86–12.04)	11.27 (10.20–12.34)	−0.013	−0.855	−0.869
Male	17.21 (16.81–17.63)	16.37 (15.96–16.78)	16.03 (15.62–16.44)	15.83 (15.39–16.26)	15.33 (14.96–15.69)	−1.187 ***	−0.705 ***	−1.892 ***
**Region**		–						
East	16.74 (16.02–17.45)	16.39 (16.05–16.74)	16.09 (15.49–16.69)	15.97 (15.35–16.59)	15.36 (14.78–15.95)	−0.652 *	−0.722 **	−1.374 ***
Central	16.93 (16.25–17.60)	16.61 (15.90–17.32)	16.37 (15.71– 17.03)	16.00 (15.54–16.47)	15.53 (15.06–15.99)	−0.555	−0.845 **	−1.401 ***
West	17.26 (16.53–17.98)	15.37 (14.45–16.28)	14.86 (14.00–15.71)	14.21 (13.71–14.71)	14.35 (13.73–14.97)	−2.398 ***	−0.508	−2.906 ***
**Residence**								
Rural	17.54 (16.98–18.08)	16.92 (16.34–17.49)	16.10 (15.53–16.66)	15.67 (15.18–16.17)	15.23 (14.72–15.73)	−1.441 ***	−0.869 ***	−2.310 ***
Urban	16.31 (15.85–16.78)	15.50 (15.07–15.93)	15.63 (15.15–16.11)	15.38 (14.79–15.96)	15.04 (14.60–15.49)	−0.681 *	−0.586 *	−1.267 ***
**Education**								
Primary or below	17.52 (16.99–18.06)	17.12 (16.50–17.74)	16.58 (15.98–17.18)	16.15 (15.47–16.82)	16.46 (15.82–17.10)	−0.945 **	−0.121	−1.066 ***
Secondary or high school	16.84 (16.28–17.40)	15.68 (15.22–16.13)	15.78 (15.31–16.24)	15.22 (14.76–15.69)	14.79 (14.37–15.22)	−1.062 ***	−0.986 ***	−2.048 ***
College or above	13.92 (13.05–14.78)	13.80 (12.95–14.64)	11.97 (11.20–12.75)	12.27 (11.51–13.03)	12.19 (11.41–12.97)	−1.943 ***	0.219	−1.724 ***
**Age, years**								
16–29	13.24 (12.39–14.09)	11.93 (11.27–12.59)	12.25 (11.73–12.77)	10.94 (10.37–11.52)	10.51 (9.98–11.04)	−0.990 **	−1.739 ***	−2.729 ***
30–39	17.08 (16.44–17.73)	16.05 (15.48–16.63)	14.77 (14.23–15.31)	14.32 (13.80–14.84)	13.97 (13.42–14.52)	−2.309 ***	−0.802 *	−3.110 ***
40–49	19.18 (18.45–19.92)	17.88 (17.32–18.45)	17.38 (16.68–18.09)	16.58 (15.98–17.18)	16.27 (15.61–16.93)	−1.800 ***	−1.117 **	−2.917 ***
50–59	18.59 (17.78–19.40)	18.75 (18.06–19.44)	18.18 (17.40–18.95)	17.90 (17.24–18.56)	17.88 (17.20–18.56)	−0.415	−0.297	−0.712
60–69	18.24 (16.88–19.61)	16.45 (15.65–17.25)	16.99 (16.04–17.95)	16.82 (15.99–17.66)	16.49 (15.83–17.15)	−1.254	−0.503	−1.757 *
≥70	13.19 (12.19–14.18)	13.39 (12.54–14.23)	11.94 (10.98–12.90)	12.69 (11.48–13.90)	12.93 (12.03–13.83)	−1.250 *	0.992	−0.257

Note: Weighted means and 90% confidence intervals are reported in the table. *T*-test was used to compare the change in means. ***, ** and * denote statistical significance at the 1%, 5%, and 10% level, respectively.

**Table 4 ijerph-19-00670-t004:** Risk factors associated with number of average daily cigarettes smoked among current smokers.

Number of Average Daily Cigarettes	Total	Women	Men
	Coef. (90% CI)	Coef. (90% CI)	Coef. (90% CI)
**Marital status**			
Unmarried			
Married or cohabited	−0.025	−1.824	0.023
	(−0.406~0.356)	(−3.714~0.066)	(−0.367~0.412)
Divorced or widowed	0.384	−1.182	0.66 *
	(−0.142~0.909)	(−3.188~0.824)	(0.103~1.218)
**Educational level**			
Primary or below			
Secondary or high school	−0.739 ***	−0.858 *	−0.932 ***
	(−0.949~−0.530)	(−1.669~−0.0478)	(−1.149~−0.716)
College or above	−2.713 ***	−1.69	−2.932 ***
	(−3.128~−2.299)	(−3.772~0.393)	(−3.356~−2.508)
**Household income**			
Lower than average			
Equal to or higher than average	0.515 ***	0.163	0.513 ***
	(0.312~0.718)	(−0.525~0.851)	(0.302~0.723)
**Work type**			
Unemployed			
Self employed	0.717 ***	−0.540	0.511 ***
	(0.442~0.991)	(−1.323~0.243)	(0.222~0.800)
Employed by others	0.469 ***	−.385	0.135
	(0.176~0.761)	(−1.537~0.767)	(−0.171~0.440)
**Current drinker**			
No			
Yes	2.064 ***	2.854 ***	1.806 ***
	(1.866~2.262)	(1.886~3.821)	(1.603~2.010)
**Physical activity**			
No			
Yes	−1.429 ***	−0.836 **	−1.473 ***
	(−1.631~−1.227)	(−1.478~−0.194)	(−1.683~−1.263)
**Observations**	43,495	2,399	41,096

Note: In columns 2 and 3, linear regression models were adjusted for geographical regions, urban/rural status, age and age square, and wave dummies using pooled data from wave of 2010, 2012, 2014, 2016, and 2018. In column 1, the linear regression model further included the gender variable in addition to other control variables. 90% Confidence intervals for coefficients were reported in the brackets. *** *p* < 0.01, ** *p* < 0.05, * *p* < 0.1.

## Data Availability

The datasets generated and/or analyzed during the current study are publicly available and could be accessible via website: http://www.isss.pku.edu.cn/cfps/ (accessed on 28 November 2021).

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
