# Peer review of "Trends in Smoking Prevalence and Intensity between 2010 and 2018: Implications for Tobacco Control in China"

_ijerph, 2022, doi:10.3390/ijerph19020670_

Round 1

Reviewer 1 Report

This is a study on the prevalence and intensity of smoking in China between 2010 and 2018, and the factors associated to those variables. The manuscript is well-written and methodologically correct. However, in my opinion, it lacks of novelty on the field since most of the descriptive results can already be retrieved from public repositories and organizations (e.g. WHO) and the associations found are largely known.

Introduction

I would recommend the authors remove the last paragraph, from the moment they mention the objective on. This excerpt includes ideas associated to methods, results and discussion.

Methods

Is there a minimum age for participating in the CFPS? Has every participant in the CFPSs been incorporated in the analysis or has someone been excluded (e.g. individuals with missing data in any variable)?

Past tense should be used throughout Methods (e.g. “we used data…”).

Variables - I would recommend the authors remove the codification of “yes/no” variables.

I see the question on intensity mentions cigarettes. Are there further data on this dataset on different tobacco products?  

Statistical analysis – perhaps the authors could add “binary” to logistic regression.

Results

Perhaps a paragraph to describe the sample would be of interest. The total sample size is not shown anywhere.

Table 1. In my opinion, “current” should be removed from the title of the table.

The authors may add as foot in the table the test used to compare the change in proportions.

Table 2. I think it would be much more informative if confidence intervals are provided instead of standard errors. N’s may be given in the headings. Authors mention in the foot that the models are adjusted for gender, but gender is also used to stratify (I assume the overall model is the one adjusted for gender).

Table 3. The authors may add as foot in the table the test used to compare the change in means.

Table 4. Again, I think it would be much more informative if confidence intervals are provided instead of standard errors.

Labels should be added to the X-axis in both graphs.

Discussion

In my opinion, there are discrepant messages in discussion on the interpretation of the results (e.g. “these results suggest China’s positive progress in tobacco control” vs. “China’s progress in tobacco control has lagged behind expectations).

Reviewer 2 Report

Title: Trends in Smoking Prevalence and Intensity between 2010 and 2018: Implications for Tobacco Control in China:

The authors use five waves of data from China Family Panel Studies (CFPS), a nationally representative survey, to examine the trends in smoking prevalence and intensity. They also explore the associations between smoking behavior with several socio-demographic variables and risk factors (using logistic regressions).

This is primarily a descriptive paper, analytically non-rigorous. I found the content or write-up a bit stretched long. It is a descriptive report of the findings from the five rounds of China Family Panel Studies (CFPS), based on two self-reported smoking-related indicators.

  1. The prevalence estimates are based on the indicator “current smoker” – is it ‘cigarette smoking’ or ‘tobacco smoking’? Please compare with the reported indicators in the WHO Global Tobacco Control reports. 
  2. The smoking intensity is measured from the survey response to the question “How many cigarettes do you currently smoke on average per day” – how was the daily average calculated for respondents who were not daily smokers?
  3. It was not clear whether the surveys are cross-sectional or longitudinal/panel. If these are panel/longitudinal – what are the attrition rates of the respondents? 
  4. Abstract: Please revise the sentence “Multiple regressions are applied to examine the association between smoking behaviors and risk factors” – the phrase “multiple regression’ is ambiguous.
  5. Abstract: “our study shows that the Chinese government has made some progress in tobacco control over the past decade. Nonetheless, tobacco control progressed more slowly than expected” – I am not sure whether this attribution, while maybe correct, can be made from this study. This study merely describes the trend in smoking prevalence and does not analyze or juxtapose the trends in various tobacco control measures by the Chinese Government with the prevalence trends. I would rather prefer a direct statement about how prevalence evolved over the past decade. 
  6. Page 2 (Introduction): “Nevertheless, the pace of reduction in prevalence and intensity is slower than expected” – what was the expectation? Is there any defined national target for reduction in smoking prevalence?
  7. Results: 3.1. Smoking prevalence: In the discussion section, the authors may discuss how these prevalence estimates compare with what is reported in various rounds of the WHO Global Tobacco Control Reports.
  8. Figure 1: The line graph is confusing. Since the figure is supposed to show age-specific trends, bar graphs (three bars for 2010/14/18 for each age group) are more suitable.
  9. Figure 2: Again, the line graph is confusing. The X-axis is not a time-trend, rather showing discrete age groups. Since the figure is supposed to show age-specific trends, bar graphs (three bars for 2010/14/18 for each age group) are more suitable.
  10. The discussion and conclusion sections are stretched long. The conclusion section is redundant.
  11. Introduction/Discussion Section: Please ensure to include any important published literature that used China family panel study involving smoking prevalence. 

Round 2

Reviewer 1 Report

I would like to thank and commend the authors for addressing all my concerns and suggestions.

I only have minor comments:

On the explanation of the codification of the variables “drinking status” and “physical activity”, what was redundant in my opinion was the codification itself (i.e. 0/1) not the categories (i.e. yes/no). I believe what should be provided in the manuscript are the categories, but not the codes.

The authors have added Poisson regression models but they are not explained in the statistical analysis.
